# Accuracy of Speech-to-Text Transcription in a Digital Cognitive Assessment for Older Adults

**DOI:** 10.3390/brainsci15101090

**Published:** 2025-10-09

**Authors:** Ariel M. Gordon, Peter E. Wais

**Affiliations:** Department of Neurology, Neuroscape and Weill Institute for Neurosciences, University of California, San Francisco, CA 94158, USA; ariel.gordon@ucsf.edu

**Keywords:** neuropsychological assessment, digital cognitive assessment, speech-to-text

## Abstract

**Background/Objectives:** Neuropsychological assessments are valuable tools for evaluating the cognitive performance of older adults. Limitations associated with these in-person paper-and-pencil tests have inspired efforts to develop digital assessments, which would expand access to cognitive screening. Digital tests, however, often lack validity relative to gold-standard paper-and-pencil versions that have been robustly validated. Speech-to-text (STT) technology has the potential to improve the validity of digital tests through its ability to capture verbal responses, yet the effect of its performance on standardized scores used for cognitive characterization is unknown. **Methods:** The present study evaluated the accuracy of Apple’s STT engine relative to ground-truth transcriptions (RQ1), as well as the effect of the engine’s transcription errors on resulting standardized scores (RQ2). Our study analyzed data from 223 older adults who completed a digital assessment on an iPad that used STT to transcribe and score task responses. These automated transcriptions were then compared against ground-truth transcriptions that were human-corrected via external recordings. **Results:** Results showed differences between STT and ground-truth transcriptions (RQ1). Nevertheless, these differences were not large enough to practically affect standardized measures of cognitive performance (RQ2). **Conclusions:** Our results demonstrate the practical utility of Apple’s STT engine for digital neuropsychological assessment and cognitive characterization. These findings support the possibility that speech-to-text, with its ability to capture and process verbal responses, will be a viable tool for increasing the validity of digital neuropsychological assessments.

## 1. Introduction

Neuropsychological assessments are valuable tools for evaluating the cognitive performance of older adults relative to population norms for their cohort. These assessments are useful for clinical diagnoses and can help detect early signs of cognitive impairment, which go undiagnosed in over half of cases [1,2]. Such early detection is crucial because it can inform specific interventions and help patients plan for future care [3,4,5,6].

People experiencing age-related decline in cognition, however, may only show subtle symptoms of diminishing capabilities in varying cognitive domains [7]. Neuropsychological assessments can reveal changes in performance in these domains, which include memory, attention, and executive function. Critically, such changes serve as the quantitative components in the clinical diagnostic process for cognitive impairment. Neuropsychological assessments also have strong utility for research purposes, as they can characterize the cognitive status of study participants. This capability allows for research into the distinct domains that evidence cognitive decline, as well as novel diagnostic or interventional tools for aging populations.

### 1.1. Digital Neuropsychological Assessment

Neuropsychological assessments have historically been administered in-clinic, by a trained clinician, with paper-and-pencil (PAP) protocols and verbal instructions over multiple hours. The advantages of these PAP protocols are their many subtests, the administrator’s ability to observe qualitative factors that may affect performance (e.g., anxiety), and robust validation of population norms developed over millions of patient observations [8,9]. However, recent advancements in the technical capabilities, ease of use, and ubiquity of personal computing devices have facilitated the development of a number of digital assessment tools that modernize how neuropsychological assessments are administered for the evaluation of older adults’ cognition [1,3,6,8,9,10,11,12,13,14,15,16,17,18].

Digital assessments offer numerous advantages over traditional PAP methods. PAP assessments are time-consuming, expensive, and emotionally trying for participants, and require live administration by a trained specialist [6,8,9,10,17,19,20]. They also entail numerous barriers to access, such as long wait times for clinic visits, needing to travel to a physical testing location, and participant anxiety towards healthcare professionals [8,10,11]. Finally, traditional PAP tests are also susceptible to variability in human administration and scoring [6,8,9]. Digital testing addresses all these limitations, achieving easier availability and access for participants, resource and time savings for clinicians or researchers, and standardization of administration and scoring [6,8,17,21]. Many digital implementations of cognitive tests have been validated in recent years, demonstrating reliable correlations with their gold-standard paper-and-pencil counterparts [1,3,6,8,12,14,15,16,17].

An important concern regarding these existing digital assessments, however, is that many lack construct validity with the PAP subtests from which they were adapted. Many standardized PAP subtests, including tests of verbal memory, lexical fluency, and semantic fluency, rely on verbal responses from the user. Yet, the majority of published digital assessments only support touch, key, or mouse input [1,3,6,11,12,13,14,15,16], and a review of 20 digital assessments published from 2015 to 2023 found only one that recorded and scored users’ verbal responses [10]. Critically, these shifts in response modality bypass the substantial cognitive demands engaged in language production. For example, when digital verbal memory tests show previously studied words on screen, this protocol evaluates participants’ recognition of displayed words versus the more demanding free recall task that is applied in PAP standardized tests. This approach lacks validity relative to PAP measures not only because it alters the psychological constructs being measured, but also because it invalidates comparisons to the standardized population norms that have been established with free-recall verbal responses.

### 1.2. Potential of Speech-to-Text for Digital Assessment

Speech-to-text (STT) technology offers a potential solution for increasing the construct validity of digital neuropsychological assessments by enabling inputs that more effectively mimic those of PAP protocols. STT technology refers to any set of algorithms that transforms linguistic audio data into text format. This capability allows digital assessments to capture and score verbal inputs, which in turn allows them to more closely mimic the methods and constructs of their gold-standard PAP analogs.

STT is straightforward to incorporate into digital assessments for two primary reasons: its ubiquity with users and its ease of implementation for developers. First, STT is highly familiar to 21st-century consumers through voice assistants like Siri, Google Assistant, and Alexa, which have been widely embedded into many modern technologies (e.g., vehicles, televisions, watches, and smartphones). This familiarity extends to older adults, who may even prefer voice interfaces over touchscreens or a mouse [22]. Second, STT engines provide software libraries (Application Programming Interfaces, or APIs) that allow developers to easily integrate these standardized tools into applications built for any modern device with a microphone and internet access.

### 1.3. Prior Work

A number of digital assessments have incorporated STT technology, and while these implementations have proven technically successful, their accuracy in measuring cognitive performance remains largely unexplored. For instance, some studies have measured STT engine accuracy relative to human transcriptions, but in applications other than neuropsychological assessment (e.g., a job interview) [23,24]. Other studies have utilized STT in such assessments to measure cognitive performance but failed to quantify its error rate against ground-truth human transcriptions [8,9].

The research that comes closest to bridging this gap in the literature is that which compared STT transcriptions to ground-truth transcriptions in verbal memory tasks, finding that the two were highly concordant [18,25,26]. However, these studies present a critical limitation. While they measured raw STT transcription accuracy (i.e., word error rate), they did not quantify the effect of these errors in terms of normative cognitive scores. Even if STT engines demonstrate high accuracy in transcribing responses, residual errors may significantly affect standardized scores relative to population norms that account for demographic criteria, which inform the characterization of a participant’s cognitive performance in a study cohort [8].

Moreover, none of these studies evaluated Apple’s native STT engine; instead, they used other providers (e.g., Google) or custom-built models [18,23,24,25,26]. This is an important practical limitation given that half of all smartphones owned by older adults in the U.S. are iPhones [27], and Apple’s native API offers the easiest integration into tools built for its devices.

### 1.4. Present Study

The present study aimed to quantify the performance of Apple’s STT API in a digital cognitive assessment for older adults, as well as interpret whether its variability may affect participants’ results relative to normative cognitive scores. These aims motivated two research questions:

Research Question 1 (RQ1): How accurate are Apple’s STT transcriptions compared to human transcriptions in a digital cognitive assessment for older adults?

Research Question 2 (RQ2): How do errors in STT transcription affect standardized scores that are used for cognitive characterization?

We addressed these questions by analyzing a dataset of older adults who completed a digital assessment developed by Arioli et al. [8]. The application transcribed verbal responses with Apple’s STT engine and used those transcriptions to tally automated task scores. These verbal responses were recorded simultaneously with a separate device, which a human coder then used to generate ground-truth transcriptions and scores. We compared the scores derived from the automated transcriptions and the manual transcriptions to quantify the accuracy of the STT engine. Then, both automated and manual scores were transformed to standardized scores with respect to participants’ demographic information. The disparity in *z*-scores was interpreted in terms of the potential for diagnostic error when using STT for neuropsychological assessment, considering that significant transcription errors could lead to the mischaracterization of an individual’s cognitive performance.

## 2. Materials and Methods

### 2.1. Participants

223 adults (109 females, 114 males) with a mean age of 70.8 years (SD = 6.3, min = 60, max = 85) were recruited via community outreach, social media advertisements, and clinical referrals coordinated by UCSF Neuroscape. Inclusion criteria included being 65 years of age or older, being a fluent speaker of English, having normal or corrected-to-normal vision and audition, and having no history of neurological or psychiatric disorders. Participants gave their informed consent in accordance with the Institutional Review Board of the University of California, San Francisco. This study was carried out in accordance with the Declaration of Helsinki, and experimental methods were carried out in accordance with the guidelines and regulations approved under UCSF IRB #19-27586.

### 2.2. RCM Digital Screener

The digital neuropsychological assessment used in the present study was an iPad application developed and validated as a cognitive screener by Arioli et al. [8]. This Remote Cognitive Module (RCM) consisted of nine tasks to assess long-term memory, working memory, verbal fluency, and attentional set-shifting.

Task 1 (T1) mimicked the California Verbal Learning Test II (CVLT-II) Immediate Word Recall task [28]. It instructed participants to listen to a list of 16 common nouns and then immediately repeat them back from memory. This was repeated five times with the same list.Task 2 (T2) was a distractor task to serve as a delay period until Task 3. It instructed participants to spell the word “WORLD” forwards, then backwards.Task 3 (T3) mimicked the CVLT-II Short Word Recall task [28]. It instructed participants to freely recall as many words as they could from the original list of 16 nouns they learned in Task 1.Task 4 (T4) mimicked the Wechsler Adult Intelligence Scale Verbal Digit Span task [29]. It instructed participants to recall an increasingly long sequence of single-digit numbers forwards, then backwards.Task 5 (T5) was a lexical fluency task [30,31] that instructed participants to say as many words that start with “B” as they can within 1 min, excluding proper nouns.Task 6 (T6) was a semantic fluency task [31] that instructed participants to say as many animals as they could within 1 min.Task 7 (T7) mimicked the Trails Making Test B task [32]. It instructed participants to draw a line on the screen to connect alternating numbers and days of the week.Task 8 (T8) mimicked the CVLT-II Long-Delay Free Recall task [28]. It instructed participants to freely recall as many words as they could from the original list of 16 nouns they learned in Task 1, after a longer delay.Task 9 (T9) mimicked the CVLT-II Cued Recall task [28]. It instructed participants to recall as many words as they could from the original list of 16 nouns they learned in Task 1, but only of a specifically cued category (e.g., furniture or vegetables). This was repeated four times, once for each category.

The version of RCM used in the present study was updated from the original version published by Arioli et al. [8]. The original version described digit span forward and backward as two separate tasks (“Task 4” and “Task 5”), and as such, the task names here are shifted from the original names. Additionally, the original version did not report results from the cued recall task.

### 2.3. Procedures

#### 2.3.1. Enrollment

All participants completed an enrollment questionnaire that confirmed they met the inclusion criteria for the study and would be able to participate either remotely or in-lab. Remote participation required participants to have a computer with a camera and microphone to support a Zoom video call, and also an iPad 10th-generation or later running iOS 14 or later. In-lab participation required participants to be willing and able to commute to our laboratory. Following the enrollment questionnaire, eligible participants e-signed an informed consent document via Qualtrics. A member of the study team then emailed a Zoom link for remote sessions or relevant logistics for in-lab sessions.

#### 2.3.2. Session Orientation

The 45 min study session began with the experimenter informing the participant that they would complete nine tasks on an iPad over 25 min, and that the iPad would provide all instructions for each task. The participant was told that the iPad uses speech-to-text technology, so they should speak clearly, concisely, and avoid saying unnecessary words or phrases (e.g., “that’s all I remember”, “um”, “darn”) to prevent errors.

The experimenter informed the participant that the study session would be recorded for quality control (QC) purposes. To collect this recording, the experimenter positioned a separate iPad or iPhone near the computer that was running the Zoom (remote) or in the room with the participant (in-lab). This device recorded the entirety of the session in the Apple Voice Memos application.

The participant entered their demographic information (age, sex, and years of education) into the first screen of the RCM application. Then, they began the assessment, which was self-guided from this point forward. During the assessment, the experimenter turned off their camera and microphone in Zoom (remote) or exited to a control room (in-lab) to observe the session, take notes, and assist with any technical difficulties.

#### 2.3.3. RCM Assessment

To begin the assessment, an introduction was displayed as text and spoken aloud by a digitized female voice, after which the participant began the first task by tapping a START button on the screen when they were ready. This process was repeated for each RCM task: instructions were both displayed and spoken, after which the participant could begin the task by pressing START. The application automatically progressed the participant through each of the nine tasks. For each task (except T7), after the participant received the instructions and pressed start, the iPad displayed a RECORDING graphic to prompt a response. The participant spoke their responses, then pressed a DONE button when they were done to advance to the next task. If a participant did not press DONE within 1 min, the task timed out and they were automatically advanced to the next task.

The original version of RCM published by Arioli et al. [8] necessitated that participants say the word “comma” in between each spoken word to aid STT processing and scoring. However, improved application of the STT API rendered this interjection no longer necessary, so it was removed to avoid confusion and distraction for participants. In the updated version, participants freely spoke each response word in sequence without saying “comma”.

#### 2.3.4. Automated STT Processing and Scoring

The six tasks of interest in the present study are those that required the participant to speak a list of words: T1 (immediate recall), T3 (short-delay recall), T5 (lexical fluency), T6 (semantic fluency), T8 (long-delay recall), and T9 (categorical recall). T2 was not of interest because it was a distractor task, and T7 (trails-making) was not of interest because it did not require verbal responses. Additionally, T4 (digit span) was not of interest because speech-to-text transcription for spoken single-digit numbers is consistently accurate due to the limited set of possible responses.

For each of these tasks, Apple’s Speech Recognition API started recording input while RECORDING was displayed on the iPad and ceased recording when either the DONE button was pressed or the task timed out. The output of this recording was then stored in the application and tokenized into individual words. The application post-processed each response to remove filler words (e.g., “um”, “sorry”), make plurals singular (e.g., “cats” → “cat”), and rectify common mistranscriptions for the list of 16 nouns (e.g., “motor cycle” → “motorcycle”, “swirl” → “squirrel”, see Appendix A for additional examples).

After initial post-processing, the application scored the spoken responses for each task. For T1 (immediate recall), T3 (short-delay recall), T8 (long-delay recall), and T9 (categorical recall), the post-processed responses were compared against the 16 target words. The first mention of any of the 16 target words was scored as a “hit”. The maximum possible score was 80 hits (16 targets × 5 runs) for T1, and 16 hits each for T3, T8, and T9. For T5 (lexical fluency), any word that started with “b” in the post-processed response was scored as a hit, excluding proper nouns. For T6 (semantic fluency), the post-processed response was compared against a dictionary of animals, and the first mention of any animal was scored as a hit.

After scoring, the application saved these automated transcriptions and scores and uploaded them to a secure server.

#### 2.3.5. Manual Transcription Correction

Following the study session, a member of the study team reviewed the automated transcriptions while listening to the QC recording. For each of the six spoken-word tasks, they corrected missing or mistranscribed words, ensuring that the corrected transcriptions reflected the true spoken responses captured in the QC recording. Then, they updated the scores for each task in accordance with the corrected transcriptions. Corrected transcriptions and scores were saved in a separate file and uploaded to a cloud storage platform, along with the QC recording and original automated transcriptions and scores. These human-corrected manual transcriptions were considered “ground-truth” transcriptions for our analyses.

## 3. Results

Our analysis had two aims, as outlined in the Introduction. First (RQ1), to quantify the accuracy of the STT transcriptions automatically generated by Apple’s API relative to ground-truth manual transcriptions. Second (RQ2), to estimate the effect of errors in STT transcription on resulting standardized scores, which are used to characterize participants’ cognitive performance with respect to their demographic cohort [8].

Four participants’ data were excluded fully from analysis: three due to errors in data capture, and one due to errors in manual correction. Additionally, some participants’ individual task data were excluded in particular cases: three participants’ T5 data were excluded because they misheard the instructions (said words that start with “D” instead of “B”), two participants’ T6 data were excluded because of errors in manual correction, and five participants’ T9 data were excluded because of a malfunction that resulted in completely missing automated transcription. All analyses were performed using RStudio Version 2023.06.1 [33].

### 3.1. Automated Versus Manual Transcriptions

To address RQ1, we compared the mean scores generated from the automated transcriptions versus the manual transcriptions, as measured in hits per task (Table 1, Figure 1a). Pairwise *t*-tests found group differences in mean scores generated from the automated transcriptions compared to mean scores generated from the manual transcriptions, and this result was true for every task (all *p*’s < 0.001). This pattern of differences shows that manual correction consistently improved scores by rectifying errors made by the STT API. These differences were unidirectional because the errors in the automated transcription (namely, missing or mistranscribed words) always resulted in lower scores relative to the participant’s true responses. As such, rectifying these errors during manual transcription correction only led to corresponding score increases.

Pairwise *t*-tests, however, only demonstrate differences between automated and manual scores at the group level. To investigate the accordance between automated and manual scores across individual participants, we also applied Pearson’s correlations, which found strong, positive relationships between automated and manual transcriptions across all tasks (all *p*’s < 0.001; Figure 1b). Correlations for each task met or exceeded *r* = 0.93 with the exception of Task 9 (*r* = 0.78). This agreement shows that although there were differences between automated and manual scores, these differences were small and consistent across each task.

### 3.2. Effect of Transcription Error on Standardized Scores

The analysis reported in Section 3.1 shows group differences and positive correlations between the STT API’s transcriptions compared to manual corrections in each task. A critical follow-up question, however, is whether these results reflect a practical difference for the purposes of cognitive characterization. To address RQ2, we analyzed whether the mean differences in raw score between the automated and the manual transcriptions (i.e., Diff_hits_) would affect a participant’s standardized score that accounts for demographic criteria. For each task, Diff_hits_ (Table 2) was calculated by subtracting the automated mean hits from the manual mean hits (Table 1).

Standardized scores were calculated by *z*-scoring each participant’s raw data with respect to normative data from their demographic cohort. These normative scores, which are organized by age, sex, and years of education, were established in the original tests with large samples [28,29,30,31] and were also demonstrated to correlate with scores on RCM’s digital adaptations [8].

At UCSF Neuroscape, these standardized RCM scores are used in the characterization of a participant’s cognitive status as “average” or “below-average”. Characterization as “below-average” depends on meeting one or more of the following criteria: (i) *z* ≤ −1.5 on any word recall task, (ii) *z* ≤ −1.5 on two or more tasks, or (iii) *z* ≤ −2.0 on any single task. These criteria were developed from our previous aging research [34,35,36] as well as accepted practices for assessing cognitive performance using PAP neuropsychological batteries [37].

To determine the STT error (i.e., Diff_hits_) required to affect standardized scores used for cognitive characterization, we established thresholds against which to compare Diff_hits_ for each task. We identified 0.5 *z* as a significant shift in a participant’s standardized score because it would have altered their designation as “average” or “below-average” according to the above criteria. For instance, if a participant’s standardized score on a word recall task was *z* = −1.5 based on automated transcription, but was then increased to −1 after manual cleaning, then they would have initially been erroneously characterized as “below-average” due to STT errors.

Accordingly, thresholds for each task were determined using the following formula:threshold*_n_* = round((SD_avg_(Task*_n_*) × 0.5), 0.5)(1)
where threshold*_n_* is the Diff_hits_ required to alter a participant’s *z*-score by 0.5 for Task *n*. SD_avg_ was the mean of all the standard deviations for a given task across each demographic grouping (age band, sex, and years of education). Thresholds were rounded to the nearest 0.5 because hits are only scored as whole numbers; therefore, any decimal value beyond 0.5 would be irrelevant for the comparison.

A comparison of Diff_hits_ to threshold_n_ shows that Diff_hits_ did not meet or exceed threshold*_n_* for any task (Table 2, Figure 2). This analysis suggests that the difference in score between the automated and manual transcriptions was not large enough to alter a participant’s *z*-score in any task to a degree that would have resulted in a meaningful change in the characterization of their cognitive performance.

### 3.3. Effect of Transcription Error on Standardized Scores, by Age Band

As an exploratory analysis, we repeated the comparison reported in Section 3.2, but stratified results by the age bands established in normative scores for each task (60–69, 70–79, or 80+). This analysis aimed to examine if discrepancies between automated and manual transcriptions varied by age band due to potentially lower skills in interfacing with the application or providing clean verbal responses, and if such discrepancies would cause meaningful changes in standardized cognitive scores.

A comparison of Diff_hits_ for each age band against each threshold_n_ found that Diff_hits_ did not exceed threshold*_n_* for any task, for any age band, except for T9 (categorical recall) in the 80+ age band (Table 3, Figure 3). These results suggest that the differences between the automated and manual transcriptions generally remained below the threshold required to meaningfully affect participants’ standardized cognitive scores and characterization of cognitive performance, regardless of age. The only exception was T9 in the 80+ age band, in which case the errors in automated transcription were significant enough to alter standardized cognitive scores by 0.5 *z* or more, on average.

## 4. Discussion

### 4.1. General Discussion

Neuropsychological assessments are important tools for measuring cognitive performance for a variety of applications. Yet, they have historically been unwieldy in their original implementation as paper-and-pencil tests requiring in-person administration by trained coordinators. The costs of these traditional methods have inspired efforts to digitize these tools, which would expand access for participants, reduce cost for researchers and clinicians, and standardize test administration. However, the design and input methods of digital tests often fail to faithfully replicate gold-standard PAP neuropsychological tests that have been robustly validated [1,3,6,10,11,12,13,14,15,16]. STT technology promises to support more faithful digitization of neuropsychological assessments through its ability to capture verbal responses with software libraries that are easy to implement in personal computing devices. Although there has been research on the accuracy of this technology, its effect on standardized scores used for cognitive characterization is unknown.

Accordingly, we measured the accuracy of Apple’s STT engine relative to human-corrected ground-truth transcriptions (RQ1), as well as the effect of the engine’s transcription errors on resulting standardized scores (RQ2). Our study analyzed data from 223 older adults, who completed digital tests of verbal memory and fluency on an iPad that transcribed and scored their responses via Apple’s STT API [8]. These automated transcriptions were then compared against ground-truth transcriptions that were generated via external recordings to assess the performance of the STT engine and its effect on standardized scores.

Our analyses found group differences and positive correlations between automated and manual scores across all tasks (RQ1). Taken together, these results suggest that transcription of verbal responses to neuropsychological tasks via Apple’s STT engine does not achieve quite the accuracy of human transcription. Nevertheless, these differences were not large enough to practically affect standardized measures of cognitive performance (RQ2). Prior research on STT accuracy has shown similar results, finding highly similar, but not exact, agreement between human and automated transcriptions [18,25,26]. Altogether, our findings demonstrate the practical utility of Apple’s STT API for digital neuropsychological assessment and cognitive characterization, which suggests that researchers and clinicians should have confidence in utilizing the technology to collect verbal assessment data.

Our findings nevertheless suggest a thoughtful approach to utilizing STT to interpret transcribed data. Our results characterize a pattern of small, but consistent errors (i.e., Diff_hits_), which indicates that it will be necessary to consider the specific task in which STT is applied. Tests with a margin of error of one or two words may lend themselves well to this application, such as the constrained verbal memory and language fluency tasks used in this battery. On the other hand, more linguistically open-ended tests, such as the Wechsler Logical Memory task [38], which requires participants to verbally recall short stories, may be more vulnerable to diminished transcription performance across greater input quantity and variety. Supporting this notion is prior research that found significant differences between automated and human transcriptions of naturalistic free-response data in both word count and linguistic features, leading the authors to conclude that STT accuracy rates were “not as high as advertised” [23] in that domain.

Regardless of assessment context, our finding that the results of RQ2 held across a range of older adults (60–80+) suggests that STT is a viable method of user input in older adult populations broadly. The one exception in these results was in the responses of the oldest age band (80+) to T9 (categorical recall), in which STT transcription errors would have been significant enough to alter characterization of their cognitive performance in this task, on average. This result is consistent with the correlation of the automated and manual scores in T9, which was relatively low (*r* = 0.78) compared to all of the other tasks (all *r*’s ≥ 0.93).

A likely explanation for these outlying T9 results could be the user experience (UX) design of the task, rather than any features of the STT API itself. In the original design of T9 by Arioli et al. [8], after pressing START on the task, users were required to press an additional button to begin recording their response. This UX contrasted with that of the other tasks, which automatically began the recording upon starting the task. Members of the research team noticed—both during data collection and while listening to the QC voice recordings—that participants often gave proper responses verbally but forgot to enable the recording button at the proper time in the task sequence, thus truncating the automated transcription. Additionally, this interaction was compounded over the four rounds of the task, one for each cued recall category. After redesigning this interaction to automatically begin recording (in service of data collected after this study), user interaction and data quality were observed to be improved, albeit anecdotally. We suggest that an implication of this finding is that deficiencies in UX may be most disruptive for the oldest users, potentially due to lower familiarity with modern technology or greater difficulties in following digital task instructions.

### 4.2. Limitations and Future Directions

The interpretations of this study are subject to several limitations. First, there were three additional types of errors that were observed during data collection: user errors, application errors, and edge cases. User errors included mishearing instructions (e.g., listing words that start with D instead of B) or saying filler words or phrases despite being instructed otherwise (e.g., “oops already said that”). Application errors included improperly tokenized words (e.g., “sea otter” transcribed as “sea” and “otter”) and incorrectly scored words (e.g., “human” not counted as an animal). These application errors were not the result of the STT API’s performance in itself, nor user confusion, but rather limitations in the utilization of the API in the application’s scoring code. Edge cases included participants who had heavy accents or who offered uncommon responses that were difficult to parse by the STT API (e.g., the animal “bontebok”). Further research is necessary on the challenges that arise when applying STT with real-world users, who may provide unexpected or irregular inputs.

Second, as described in the Methods, the automated STT transcriptions underwent a post-processing step that rectified common mistranscriptions for the list of 16 nouns in T1, T3, T8, and T9 (e.g., “motor cycle” → “motorcycle”, “swirl” → “squirrel”, see Appendix A for additional examples). This was because the primary application of the RCM tool by Arioli et al. [8] is to screen and characterize participants for study recruitment, rather than to assess STT performance. Accordingly, their priority was to ensure accurate results that were faithful to participants’ true responses, rather than to capture the raw output of the STT API. These post hoc fixes of the automated transcriptions applied by Arioli et al. [8] may have augmented the STT API’s performance in the results for those tasks. These corrections, however, were unlikely to augment the sum of hits from a participant’s responses. This is because there were only 16 target responses for these tasks, all with distinct meanings and pronunciations, which makes it reasonable to assume a participant would not have intended to say irrelevant close phonetic matches (e.g., “swirl” rather than “squirrel”).

Third, there were also limitations in the application’s post-processing of transcriptions on one task. Specifically, when scoring responses for the semantic fluency task (T6), the code did not account for compound animal names nor every animal name in the English language (US). This limitation might have led to underestimations of the STT API’s performance on T6. More sophisticated tokenization of responses to account for compound nouns and a comprehensive scoring dictionary could yield a truer assessment of STT’s accuracy on this open-ended fluency task.

Still, scoring words via comparison to a static list may be an inferior approach to using large language models (LLMs). Future research should examine post-transcription scoring with LLM APIs that can evaluate transcribed words in the context of embedded linguistic knowledge rather than a list of pre-selected correct answers. Other directions for future research include evaluating the newest STT models, which may demonstrate superior performance due to recent advances in machine learning architecture.

## 5. Conclusions

The present study measured the performance of Apple’s STT API in a digital cognitive assessment for older adults and found that while automated transcriptions did deviate from human transcriptions, this difference was not enough to have practical effects on the characterization of cognitive performance. Novel findings of this work include the specific evaluation of Apple’s API and the analysis of the effect of STT transcription errors on standardized cognitive scores. These results support the possibility that speech-to-text, with its ability to capture and process verbal responses, will be a viable tool for increasing the validity and utility of digital neuropsychological assessments.

## Figures and Tables

**Figure 1 brainsci-15-01090-f001:**
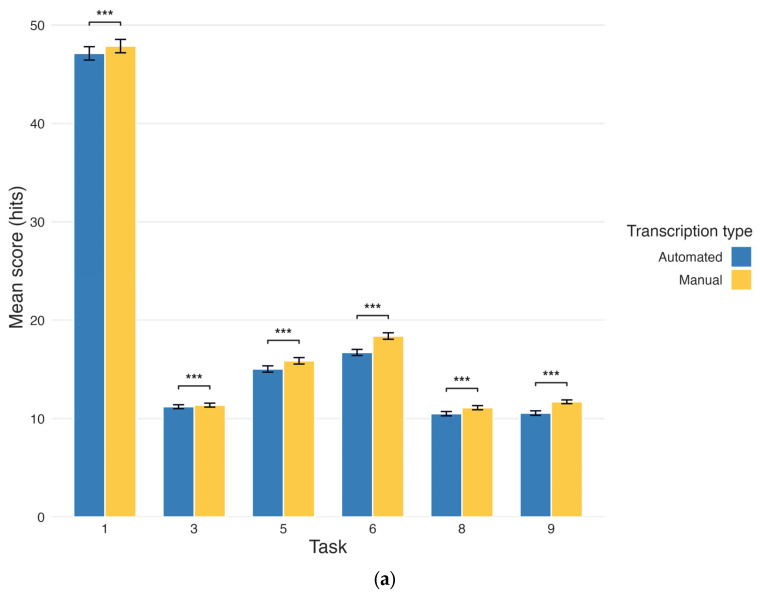
(**a**) Mean score per task for automated and manual transcriptions. The tasks shown are: 1 (immediate recall), 3 (short-delay recall), 5 (lexical fluency), 6 (semantic fluency), 8 (long-delay recall), and 9 (categorical recall). Blue and yellow bars represent the mean number of hits for each task based on automated and manual transcriptions, respectively. Error bars indicate the standard error of the mean (SEM). *** indicates a significant difference, *p* < 0.001. (**b**) Correlation between automated and manual scores. The tasks shown are: 1 (immediate recall), 3 (short-delay recall), 5 (lexical fluency), 6 (semantic fluency), 8 (long-delay recall), and 9 (categorical recall). Points represent an individual participant’s score, with higher opacity indicating multiple participants with the same score. Dashed red lines represent the best linear fit across the data. Pearson’s correlation coefficient with degrees of freedom (*r*(df)) and *p*-values are displayed for each task.

**Figure 2 brainsci-15-01090-f002:**
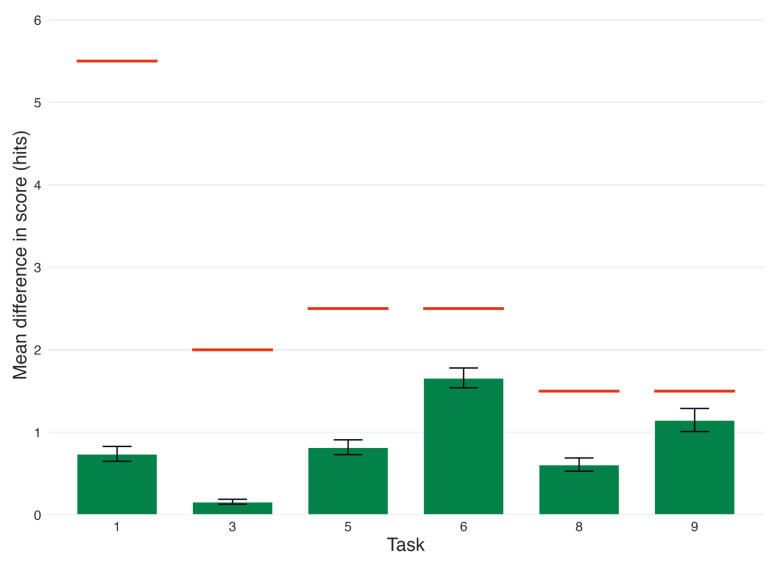
Mean difference in score per task, compared to thresholds. The tasks shown are: 1 (immediate recall), 3 (short-delay recall), 5 (lexical fluency), 6 (semantic fluency), 8 (long-delay recall), and 9 (categorical recall). Green bars represent the mean difference in hits (i.e., Diff_hits_) between automated and manual transcriptions for each task. Red lines represent the Diff_hits_ required to alter a participant’s *z*-score by 0.5 for Task *n* (i.e., threshold*_n_*). Error bars indicate the standard error of the mean (SEM).

**Figure 3 brainsci-15-01090-f003:**
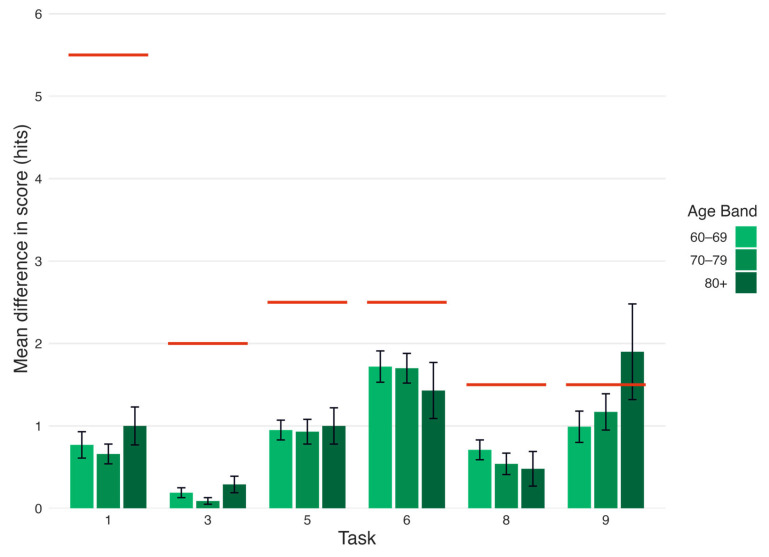
Mean difference in score per task, by age band, compared to thresholds. The tasks shown are: 1 (immediate recall), 3 (short-delay recall), 5 (lexical fluency), 6 (semantic fluency), 8 (long-delay recall), and 9 (categorical recall). Green bars represent the mean difference in hits (i.e., Diff_hits_) between automated and manual transcriptions for each task, with the shade of green indicating the age band. Red lines represent the Diff_hits_ required to alter a participant’s *z*-score by 0.5 for Task *n* (i.e., threshold*_n_*). Error bars indicate the standard error of the mean (SEM).

**Table 1 brainsci-15-01090-t001:** Mean score per task for automated and manual transcriptions.

Task	Automated Score	Manual Score	Pairwise *t*-Test
	Mean Hits (SEM)		
1 Immediate recall	47.13 (0.68)	47.87 (0.68)	*t*_218_ = 7.74, *p* < 0.001
3 Short-delay recall	11.21 (0.20)	11.37 (0.20)	*t*_218_ = 4.59, *p* < 0.001
5 Lexical fluency	15.04 (0.32)	15.87 (0.33)	*t*_215_ = 8.93, *p* < 0.001
6 Semantic fluency	16.72 (0.31)	18.39 (0.33)	*t*_216_ = 13.40, *p* < 0.001
8 Long-delay recall	10.50 (0.22)	11.11 (0.21)	*t*_217_ = 7.50, *p* < 0.001
9 Categorical recall	10.56 (0.23)	11.71 (0.19)	*t*_213_ = 8.15, *p* < 0.001

**Table 2 brainsci-15-01090-t002:** Mean difference in score per task, compared to thresholds.

Task	Diff_hits_ (SEM)	Threshold*_n_* (Hits)	Diff_hits_ > Threshold*_n_*	*n*
1 Immediate recall	0.74 (0.09)	5.5	False	219
3 Short-delay recall	0.16 (0.03)	2.0	False	219
5 Lexical fluency	0.82 (0.09)	2.5	False	216
6 Semantic fluency	1.66 (0.12)	2.5	False	217
8 Long-delay recall	0.61 (0.08)	1.5	False	219
9 Categorical recall	1.15 (0.14)	1.5	False	214

**Table 3 brainsci-15-01090-t003:** Mean difference in score per task, by age band, compared to thresholds.

Task	Diff_hits_ (SEM)	Threshold*_n_* (Hits)	Diff_hits_ > Threshold*_n_*	*n*
60–69				
1 Immediate recall	0.77 (0.16)	5.5	False	101
3 Short-delay recall	0.19 (0.06)	2.0	False	101
5 Lexical fluency	0.95 (0.12)	2.5	False	101
6 Semantic fluency	1.72 (0.19)	2.5	False	99
8 Long-delay recall	0.71 (0.12)	1.5	False	101
9 Categorical recall	0.99 (0.19)	1.5	False	99
70–79				
1 Immediate recall	0.66 (0.12)	5.5	False	97
3 Short-delay recall	0.09 (0.04)	2.0	False	97
5 Lexical fluency	0.93 (0.15)	2.5	False	94
6 Semantic fluency	1.70 (0.18)	2.5	False	97
8 Long-delay recall	0.54 (0.13)	1.5	False	97
9 Categorical recall	1.17 (0.22)	1.5	False	95
80+				
1 Immediate recall	1.00 (0.23)	5.5	False	21
3 Short-delay recall	0.29 (0.10)	2.0	False	21
5 Lexical fluency	1.00 (0.22)	2.5	False	21
6 Semantic fluency	1.43 (0.34)	2.5	False	21
8 Long-delay recall	0.48 (0.21)	1.5	False	21
9 Categorical recall	1.90 (0.58)	1.5	True	21

## Data Availability

The data presented in this study are available on request from the corresponding author due to data privacy requirements defined in UCSF IRB #19-27586.

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
