# Peer review of "Accuracy of Speech-to-Text Transcription in a Digital Cognitive Assessment for Older Adults"

_brainsci, 2025, doi:10.3390/brainsci15101090_

Round 1

Reviewer 1 Report

Comments and Suggestions for Authors

The authors present useful and timely research on the concurrent validity of a speech-to-text scoring method of recorded verbal responses to neuropsychological tests compared to manually scoring them. The results show small but significant differences across the board that do not surpass a 0.5 z score threshold established to classify meaningful or practical effects. Importantly, this study sets expectations for the accuracy of comercially available STT technology on a range of different neuropsychological tests. The study is interesting, useful, and timely describing the application of accessible vanguard technology.

However, the current statistical approach for RQ1 needs to be improved to show not only group level differences but also individual level agreement between measures. Additionally, a bigger effort needs to be made to justify the selected threshold for meaningful or practical differences in RQ2. Below I provide detail for these two major issues and finally, line-by-line revisions  

1. T-tests are insufficient for RQ1

RQ1 is addressed with t-tests which only address differences at the group level but tell us nothing about individual differences. Is this decrease in scores for STT uniform across the sample? This would allow the application of a correction factor to increase concurrent validity with the manual transcriptions. What do the distributions look like? Figure 1 is very adecuate to represent the t-test analysis but has the same shortcomings. A typical approach that is sensitive to individual differences that is also referenced in the introduction here is using correlation, and perhaps the most adecuate approach would be to show the 95% limits of agreement (LoA) between both measures (Bland and Altman, 1999).

LoA analysis compares the mean between measures (plotted in the x-axis of a scatter-plot) against their differences (plotted in the y-axis). The mean difference (y) is represented in the bias metric (drawn horizontally in the plot), and its difference against zero (one-sample t-test) is equivalent to the presented analysis (pairwise t-tests). Additionally, from the standard deviation of the differences between measures, LoA gives a representation of where 95% of the differences between measures are expected to lie. LoA is a visual analysis that affords one to assess the representativeness of the summary information for the individual cases, that is across different levels of performance.

Bland, J. M., & Altman, D. G. (1999). Measuring agreement in method comparison studies. Statistical Methods in Medical Research, 8(2), 135–160. https://doi.org/10.1191/096228099673819272

2. The threshold for meaningful change requires further explanation

The selection of 0.5 z as the “magnitude that could represent a meaningful difference” requires further explanation (Lines 311-314). Meaningful for what? Otherwise, this choice may seem inadequate for some cases (see Funder & Ozer, 2019). In that same article, the authors recommend the use of benchmarks to communicate the size of an effect. I wonder how the magnitudes of the selected thresholds (presented in table 2 as thresholds) compare to the magnitudes of influence estimated to standardize scores on each test accounting for age or years of education. Using benchmarks authors could more clearly describe “meaningful” effect sizes here.

Please consider a more continuous approach for RQ2 instead of this dichotomical decision around an arbitrary threshold. Small differences across all tasks relative to established relevant factors (age, years of education) can still be related this way, but perhaps it is more useful to describe to describe the interaction with the type of task. This is done in lines 391-399 where an interesting discussion of the relative differences (STT vs manual) across tasks is provided.  

Funder, D. C., & Ozer, D. J. (2019). Evaluating Effect Size in Psychological Research: Sense and Nonsense. Advances in Methods and Practices in Psychological Science, 2(2), 156–168. https://doi.org/10.1177/2515245919847202

3. Line-by-line revision

Line 83. Perhaps some of the problem. There is still a social structural component absent in digital self-assessment application. Also, there are differences in the observed data here that indicate deviations arise with STT vs traditional PAP protocols in scoring the tests. I would recommend avoiding this claim.

Line 102-103. It is not immediately clear to me why are naturalistic settings not relevant to the neuropsychological assessment under study here. A lot of the cost reductions mentioned above are only possible if digital assessment is conducted remotely, that is, in naturalistic settings.

Line 149-150. Limitations better be discussed excluding older adults with hearing/visual difficulties.

Line 202. Monitored sessions although a nice quality control, impact the scalability of the observations to remote settings self-assessment

Line 219. Were participants wearing headphones? Monitored

Line 239. It reads to me as an interesting assessment to show as a demonstration that STT can be fully accurate.

Line 249. I’m curious as to the rectification of common mistranscriptions, perhaps this is useful supplemental information. Were there any cases where STT rectified an incorrect response (e.g. the person actually said “swirl” in the ground truth evaluation)? Rectification occurred for what common mistranscriptions? Could they be listed?

Line 268. Were these manually corrected scores the ones used as “ground-truth”? If so, perhaps this could be indicated here explicitly.

Line 325 (Table 2). Most of the information of this table can be inferred from the one before. Only the threshold (hits) column presents new information.

Line 326 (Figure 2). It seems hits and score are used interchangeably here. If so, please use only one term, otherwise it makes the reader wonder about details that are not presented in the methods (i.e. how are scores calculated on each task).

Line 350 (Figure 3). Just wanted to mention that the age bands represented here are compatible with the required scatter-plots for LoA as they can be color coded in this way. Also, that the selected threshold for each task can be represented in the LoA plots in addition to the bias, cero, and the limits of agreement. Thus, a single figure (with 9 subpanels, one for each task) could replace the present Figures 1-3 in terms of information displayed.

Line 388-399. I think the matter discussed here is not context but task type. An example of context would be at-home vs in-lab.

Line 409-410. Odd sentence, revise wording.

Reviewer 2 Report

Comments and Suggestions for Authors

In the article entitled “Accuracy of speech-to-text transcription in a digital cognitive assessment for older adults,” the authors evaluated digital neuropsychological tests in older adults using speech-to-text technology. A rigorous design was used, with a sample of 223 older adults performing digitally validated tasks, in contrast to automatic transcriptions with manual corrections.

There are some points that are important to address.

  1. The automated transcription system was modified with correction rules, which mitigates STT errors but alters the assessment of its actual performance.
  2. Semantically complex tasks such as T6 and T9 are affected by tokenization errors and a limited dictionary. This could underestimate the difficulty of automatic processing in real contexts.
  3. A standard STT error metric such as WER (word error rate) was no used, which prevents direct comparison with other STT studies.

4.Several participants were excluded due to application errors, and potential bias was no assessed.

Suggestions.

  1. It is recommended to calculate metrics such as WER and sentence error rate for quantitative and comparative evaluation.
  2. It could be explored how differences between transcripts would affect not only z-scores, but also specific clinical decisions, such as diagnostic thresholds for mild cognitive impairment or dementia.
  3. Tools such as Large Language Models could be integrated to improve transcription, such as semantic classification.

The system interface could be improved in terms of user experience, such as T9, where unlike other tasks, it did not start recording automatically after pressing “START.” Participants also had to press an additional button to start recording, which many forgot to do.

Round 2

Reviewer 1 Report

Comments and Suggestions for Authors

I am satisfied overall with the work put into this revised manuscript and more specifically with the responses to the issues I raised. I have no further revisions. Good work!

Reviewer 2 Report

Comments and Suggestions for Authors

The authors have submitted a revised version (V2) of their manuscript, which demonstrates significant improvements over the initial submission (V1). The revisions are primarily conceptual and editorial, but they effectively enhance the clarity, accuracy, and framing of the study. Importantly, the scientific content, data, and results remain consistent, ensuring that the validity of the findings is preserved.

Main Improvements in V2

Corresponding Author Update

The corresponding author has been changed from Ariel M. Gordon to Peter E. Wais. This adjustment ensures accuracy in authorship responsibilities and correspondence management.

Conceptual Reframing

The manuscript has shifted from emphasizing “diagnostic evaluation” to “cognitive characterization.”

This is a key improvement because it prevents overstatement of clinical implications and aligns the study with its true scope—namely, research-oriented cognitive assessment rather than immediate diagnostic application.

Abstract and Introduction Revisions

The language in the abstract and introduction has been refined. Phrases such as “used for diagnostic evaluation” have been revised to “used for cognitive characterization.”

These changes improve precision, reduce ambiguity, and set a more balanced tone for the paper.

Discussion Refinement

The discussion in V2 is more measured and careful in describing the implications of the results. Instead of presenting the findings as directly relevant to clinical diagnosis, the revised text highlights their utility for research, cognitive screening, and potential future applications.

This approach increases the manuscript’s credibility and avoids overstated claims.

Editorial and Stylistic Enhancements

The revised version contains smoother transitions, more concise phrasing, and a more professional presentation overall.

Updated contact information and formatting corrections further improve the manuscript’s quality.

Elements that Remain Strong

Data and Results: All tables, figures, and statistical analyses remain unchanged and continue to provide strong evidence for the study’s conclusions.

Key Findings: Both versions confirm that Apple’s speech-to-text engine introduces small transcription errors but that these errors do not significantly affect standardized scores—with the exception of categorical recall in adults aged 80+, where discrepancies may matter.

Scientific Contribution: The manuscript contributes valuable knowledge on the feasibility of using speech-to-text technologies in digital neuropsychological assessment, addressing an important gap in the literature.

Overall Recommendation

The revised manuscript represents a clear improvement over the original version. By refining terminology, improving the framing of results, and enhancing editorial quality, the authors have produced a more rigorous and balanced paper.